# Diet Quality Affects the Association between Census-Based Neighborhood Deprivation and All-Cause Mortality in Japanese Men and Women: The Japan Public Health Center-Based Prospective Study

**DOI:** 10.3390/nu11092194

**Published:** 2019-09-12

**Authors:** Kayo Kurotani, Kaori Honjo, Tomoki Nakaya, Ai Ikeda, Tetsuya Mizoue, Norie Sawada, Shoichiro Tsugane

**Affiliations:** 1Department of Nutritional Epidemiology and Shokuiku, National Institute of Health and Nutrition, National Institutes of Biomedical Innovation, Health and Nutrition, Tokyo 162-8636, Japan; 2Department of Epidemiology and Prevention, National Center for Global Health and Medicine, Tokyo 162-8655, Japan; mizoue@hosp.ncgm.go.jp; 3Psychology and Behavior Sciences, Osaka Medical College, Osaka 569-0801, Japan; khonjo@osaka-med.ac.jp; 4Graduate School of Environmental Studies, Tohoku University, Miyagi 980-8577, Japan; tomo.nakaya@gmail.com; 5Department of Public Health, Juntendo University School of Medicine, Tokyo 113-0033, Japan; a-noda@juntendo.ac.jp; 6Epidemiology and Prevention Group, Research Center for Cancer Prevention and Screening, National Cancer Center, Tokyo 104-0045, Japan; nsawada@ncc.go.jp (N.S.); stsugane@ncc.go.jp (S.T.)

**Keywords:** diet quality, neighborhood deprivation, Japanese areal deprivation index, neighborhood socioeconomic status, hazard ratios, mortality, Japanese Food Guide Spinning Top, well-balanced diet, early death

## Abstract

Background: Individuals residing in more deprived areas with a lower diet quality might have a higher mortality risk. We aimed to examine the association between deprivation within an area and all-cause mortality risk according to diet quality. Methods: We conducted a population-based prospective study on 27,994 men and 33,273 women aged 45–75 years. Neighborhood deprivation was assessed using the Japanese areal deprivation index (ADI). Dietary intakes were assessed using a validated 147-item food frequency questionnaire. Results: Individuals residing in the most deprived area had the lowest dietary scores. During the 16.7-year follow-up, compared to individuals with a high quality diet residing in the least deprived area, individuals with a low quality diet had a higher risk of mortality according to increment of ADI (*p* trend = 0.03); the multivariate-adjusted hazard ratio (95% confidence interval) was 1.09 (0.999–1.19), 1.17 (1.08–1.27), and 1.19 (1.08–1.32) in those residing in the lowest through the highest third of ADI, respectively. However, individuals with a high quality diet had no significant association between ADI and mortality. Conclusion: A well-balanced diet may prevent early death associated with neighborhood socioeconomic status among those residing in highly deprived areas.

## 1. Introduction

Interest in the association between socioeconomic inequalities and health status has arisen [1]. Recently, several studies reported that neighborhood socioeconomic status have been associated with cardiovascular disease and cancer [2,3,4,5,6,7]. Prospective studies in U.S., Sweden, and France, and Japan reported that people residing in neighborhoods within more deprived areas show increased risk of all-cause mortality [3,4,5,8]. Additionally, a systematic review showed that people residing in neighborhoods within more deprived areas intake lower amounts of fruits and vegetables [9]. Thus, people residing in neighborhoods within more deprived areas may have a lower diet quality, which is a lower adherence to the dietary guidelines, and is at higher risk of mortality.

Several studies showed that individuals with a lower diet quality had a higher risk of mortality [10,11,12,13]. According to the report from the Global Burden of Diseases, an unhealthy diet was an important ‘modifiable’ behavioral risk factor for many health conditions [14]. Improvement of diet quality might play a role in the reduction of health disparities resulting from neighborhoods deprivation level.

Diet quality of individuals differs even if those who reside in the same deprived area. We hypothesized that diet quality as a moderator of the association between area deprivation and mortality. To our knowledge, no study has examined the association between the level of deprivation in an area and mortality according to diet quality. Here, we prospectively examined the association between deprivation in an area and all-cause mortality risk according to diet quality with reference to the Japanese Food Guide Spinning Top, which was jointly developed by the Ministry of Health, Labor, and Welfare and the Ministry of Agriculture, Forestry, and Fisheries [15].

## 2. Methods

### 2.1. Study Design

The Japan Public Health Center-based Prospective (JPHC) Study was launched in 1990 for cohort 1, and in 1993 for cohort 2 [16]. The participants of cohort 1 included residents of 5 Japanese public health center areas aged 40–59 years. The participants of cohort 2 included residents of 6 public health center areas aged 40–69 years. Using a self-administrated questionnaire, we collected data on medical histories and health-related lifestyles at baseline, and at five- and ten-year follow-ups. As detailed information on food intake was available at the five-year follow-up survey, we regarded the five-year follow-up survey as the baseline of the present analysis. We informed potential participants of the study objectives, and we reasoned that those who returned the study questionnaire consented to the survey participation. This study was approved by the Institutional Review Board of the National Cancer Center of Japan, and the Ethics Committee of National Institute of Health and Nutrition, Japan. The procedures followed were in accordance with the Helsinki Declaration of 1975 as revised in 1983.

### 2.2. Measurement of Area Deprivation

We used geographic information on the address of each participant at baseline. Nakaya et al. created a Japanese census-based deprivation index [8,17]. We calculated the areal deprivation index (ADI) based on deprivation-related census variables in each *chocho-aza* (CA) unit by using the small area statistics of the 1995 population census of Japan. This procedure is similar to that of the Breadline Britain poverty measure [18] and European transnational ecological deprivation measure [19,20]. We constructed an ADI for each 489 CAs. The median area was 2.41 km^2^, median population was 1176, and median number of households was 334.

### 2.3. Food Frequency Questionnaire and The Japanese Food Guide Spinning Top Score

We used data from the five-year follow-up survey that included 147 food and beverage items and nine frequency categories [21]. The food frequency questionnaire has reasonable validity and reproducibility [22,23,24].

We referred to the Japanese Food Guide Spinning Top, a chart designed for the general public indicating recommended daily servings for some food groups, with illustrations featuring examples of foods and dishes that meet the recommendations, using data from a large-scale, population-based, cohort study in Japan, for our assessments. The Japanese Food Guide Spinning Top defines food servings for each food category as follows [15]: One serving of grains is approximately 40 g of carbohydrates, one serving of vegetables weighs approximately 70 g (uncooked), one serving of fish and/or meat is approximately 6 g of protein, one serving of milk is approximately 100 mg of calcium, and one serving of fruits weighs approximately 100 g. Cases with 100% vegetable juice and 100% fruit juice are counted as half the weight of the amount actually consumed. The recommended serving sizes for each food category and the recommended total energy intake are specified according to sex, age, and physical activity level; whereas, the amount of energy intake from snacks and alcoholic beverages is recommended to be <200 kcal/day for all subgroups. We determined scores by measuring adherence to the Japanese Food Guide Spinning Top based on data from the dietary questionnaire. The procedure of creating an adherence score for the Japanese Food Guide Spinning Top has been described elsewhere [11]. We assigned an adherence score (between 0 and 10) for each of items, and calculated the total score, ranging from 0 (the lowest diet quality) to 70 (the highest diet quality), by summing the scores of all dish groups. 

### 2.4. Study Subjects

Of the potential subjects at baseline (*n* = 140,420), two public health center areas in metropolitan Tokyo and Osaka were excluded (*n* = 23,524) because the data sampling methods in those two areas were different from other areas. Of the remaining 116,896 participants, we excluded 4,378 participants, due to ineligibility, and 23,547 participants because no census information was provided by the statistical bureau for their *chocho-aza*. A total of 77,453 participants from the remaining 88,971 participants responded to the five-year follow-up survey; of these, 76,678 completed the food frequency questionnaire at the five-year follow-up survey. We further excluded 7977 participants, due to the lack of information on the number of rice bowls consumed, or the frequency of intake of more than half of each dish item, or all snack or alcoholic beverage items. Of the remaining 68,701 participants, we excluded 4668 participants with the extreme dietary intake (the upper 1% of the sex-specific intake of each dish category or either the upper or lower 1% of the sex-specific intake of total energy). Additionally, 2766 participants who reported a history of cancer, stroke, ischemic heart disease, or chronic liver disease in either the baseline or five-year follow-up survey were excluded. Ultimately, 61,267 participants (27,994 men and 33,273 women) remained for the present analysis (Appendix A). 

### 2.5. Follow-up and Outcome

The residency and vital status of the study participants were confirmed from the five-year follow-up survey through 31 December 2014, using the residential registry. Causes of deaths were ascertained via death certificates (with permission), and coded according to the ICD-10 [25]. The major endpoint of the present study was all-cause mortality. 

### 2.6. Statistical Analysis

We calculated person-years of follow-up for each person starting from the date of response to the five-year follow-up survey questionnaire until the date of death, emigration from Japan, from the public health center areas, or from the study areas in the same public health center areas, or 31 December 2014, whichever came first. For those who were lost to the follow-up, we used the last confirmed date of residence in the study area as the censoring date. Participants were divided into tertiles of ADI. We adjusted for the following variables: Age (years, continuous), sex, study area (9 areas), population density (person/km^2^, quartiles), body mass index (BMI <21, 21.0–22.9, 23.0–24.9, 25.0–26.9, ≥27.0 or kg/m^2^), smoking status (lifetime non-smoker, former smoker, or current smoker), total physical activity (metabolic equivalent task h/day, quartiles), a history of diabetes mellitus (yes or no), a history of hypertension (yes or no), a history of dyslipidemia (yes or no), coffee consumption (almost never, <1, 1, or ≥2 cups/day (1 cup = 120 mL)), green tea consumption (almost never, <1, 1, 2–3, or ≥4 cups/day), occupation (agriculture/forestry/fishery, salaried/professional, self-employed, multiple occupations, housework/unemployed, or other), and living alone (yes or no). Age- and sex- adjusted mean, the proportion of characteristics, and intakes of dietary factors were calculated according to ADI using linear or logistic regression. Data were stratified by diet quality: Individuals with median or higher scores were classified into a high quality diet group and the remaining into a low quality diet group. We imputed missing data using multivariate normal imputation, with five rounds of imputations that included all covariates, follow-up length and mortality status to account for missing data on BMI, smoking status, physical activity, coffee consumption, green tea consumption, or occupation. We combined the risk estimates of each imputed dataset using the Rubin’s rules [26]. We used Cox proportional hazards regression analysis to estimate hazard ratios (HR) and 95% confidence intervals (CI) of all-cause mortality for tertiles of ADI by diet quality level. We also calculated HRs and 95% CIs of all-cause mortality for combinations of ADI and diet quality when the lowest ADI and high diet quality subgroup was taken as reference. The model was adjusted for age, sex, study area, population density, BMI, smoking status, total physical activity, a history of diabetes mellitus, a history of hypertension, a history of dyslipidemia, coffee consumption, green tea consumption, occupation, and living status. Interactions between ADI and diet quality were examined by including the respective interaction terms in the model. We tested the proportional hazards assumption by including a product term between tertiles of ADI and the follow-up period in the models; no significant violation of the assumption was found (all *p*-values > 0.10). All analyses were performed using SAS version 9.4 for Windows (SAS Institute, Cary, NC, USA).

## 3. Results

Characteristics of the study participants, according to ADI, are shown in Table 1. The mean (standard deviation) ADI were 478.8 (35.2), 549.5 (19.7), and 664 (66.1) in individuals residing in the least deprived area through the most deprived area, respectively. Individuals residing in more deprived areas were more likely to be old, female, with higher BMI, physically active, engaged in agriculture, forestry, and fishery, and to live alone. In contrast, they were less likely to be a current smoker and have habitual alcohol consumption.

Individuals residing in the most deprived area had the lowest scores on the adherence to the Japanese Food Guide Spinning Top; the age- and sex-adjusted mean (standard error) scores were 48.3 (0.06), 48.5 (0.06), and 46.6 (0.06), (*p* < 0.0001 for trend) in individuals residing in the least deprived area through the most deprived area, respectively (Table 2). Individuals residing in more deprived areas had lower amounts of intakes of grains, potatoes, vegetables, fruits, mushrooms, fish and shellfish, dairy products, and green tea, whereas, they had higher amounts of intakes of pulses, meat, and coffee. They had lower intake amounts of total energy, protein, cholesterol, carbohydrate, dietary fiber, calcium, and sodium, while they consumed higher amounts of fat.

During the mean follow-up time of 16.5 years, 10072 participants died. Figure 1 shows that individuals with a low quality diet had a higher risk of mortality according to increment of ADI (*p* = 0.03 for trend), compared with individuals with a high quality diet residing in the least deprived area; the multivariate-adjusted HRs (95% CI) were 1.09 (0.999–1.19), 1.17 (1.08–1.27), and 1.19 (1.08–1.32) for those residing in the lowest through the highest third of ADI, respectively. However, individuals with a high quality diet had no significant association between ADI and mortality (*p* =0.92 for trend, *p* = 0.26 for interaction with diet quality); the multivariate-adjusted HRs (95% CI) were 1.00 (reference), 1.01 (0.93–1.10), and 1.05 (0.96–1.16) in those residing in the lowest through the highest third of ADI, respectively.

## 4. Discussion

In this large prospective cohort in Japan, individuals residing in the more deprived area had a low diet quality. Individuals residing in the highest third of ADI had a low quality diet, and a 19% higher total mortality risk compared to individuals residing in the lowest third of ADI with a high quality diet, whereas, individuals with a high quality diet had no significant association between ADI and total mortality. To the best of our knowledge, the present study is the first study to examine the association between ADI and mortality according to diet quality.

We found that individuals residing in the more deprived area had not only low intakes of fruits and vegetables, but also low intakes of fish and shellfish, and dairy products and high intake of meat. Similarly, a systematic review showed that people residing in neighborhood within more deprived areas had lower amounts of fruits and vegetables in their diets [9]. We may consider individuals residing in more deprived areas to be suffering from “food deserts” [27], areas of poor access to healthy affordable food in which the population is characterized by deprivation and compound social exclusion. A systematic review from 38 papers showed that better food access (availability, accessibility, affordability, accommodation, and acceptability) was associated with better diet quality [28]. Furthermore, in deprived neighborhoods, those who have a poor diet might be more seriously associated with poverty and social isolation (i.e., eating alone) compared to those who have a healthy diet [29]. It might be beneficial to improve both food and social access to healthy foods, such as vegetables, fruits, fish, and shellfish among individuals residing in the more deprived areas for reduction of diet disparities.

Several studies showed that individuals residing in a more deprived area had a higher risk of all-cause mortality [3,4,5,8]. However, we observed that the risk of all-cause mortality had different associations with ADI between low and high diet quality subgroups. Among individuals with a higher quality diet, we found no clear association between ADI and all-cause mortality. In contrast, among low quality diet subgroups, we found that a more deprived area was associated with a higher risk of mortality. To date, high diet quality, such as greater adherence to the Dietary Guidelines for Americans or the Mediterranean diet, has been associated with a lower risk of total or cause-specific mortality [10,11,12]. In the JPHC Study, Kurotani and colleagues previously reported that a higher adherence to the Japanese Food Guide Spinning Top was associated with a lower risk of total or cause-specific mortality [11]. Whereas, the causes of the difference in the association between ADI and mortality by diet quality were unclear, the diet quality might reflect other risk factors. We found that individuals with lower quality diet tended to be current smokers and living alone. However, we adjusted for smoking and living status in the main analysis, suggesting that differential association by diet quality may not be ascribed to these factors. It suggests that a high quality diet might have the effect of decreasing the risk of mortality beyond that of increased risk of mortality associated with area deprivation.

The strengths of the present study are its population-based prospective design involving a large cohort derived from multiple areas across Japan, the long duration of follow-up (mean 16.7 years), and the use of a validated food frequency questionnaire. This study also has several limitations. First, we excluded two metropolitan areas (Tokyo and Osaka). The present findings might not be applicable to populations in these areas. Second, we could not update the neighborhood deprivation status of the present study population because of the data available for constructing area factors [8]. However, the area socioeconomic characteristics did not dramatically change in Japan, especially in non-metropolitan areas. The proportion of emigration from the public health center areas or the study areas in the same public health center areas was 11% in non-metropolitan areas, whereas, that in metropolitan areas was 24% in this study. Third, we could not consider emigration within the study areas in the same public health center areas, although the date of the emigration was used as the censoring date. This misclassification might lead to a null association. Fourth, we only used data on dietary intake assessed at the five-year follow-up survey. However, dietary intake was generally stable over time; the Spearman rank correlation coefficients of each dish category intake between the five-year and the ten-year follow-up surveys ranged between 0.46 and 0.64 in men, and between 0.45 and 0.64 in women [11]. Finally, the effects of confounding by residual and unmeasured variables cannot be completely ruled out. Unfortunately, we did not have sufficient information on individual-level socioeconomic conditions. Specifically, we adjusted for each participant’s occupation in a broadly-defined category, which cannot precisely capture individual socioeconomic status. 

## 5. Conclusions

In summary, high diet quality was associated with a lower risk of all-cause mortality regardless of area deprived levels, whereas, low diet quality was associated with a higher risk of all-cause mortality according to area deprivation. Our findings suggest that a well-balanced diet, such as closer adherence to the Japanese Food Spinning Top, can contribute to reduced health disparities. Further research is needed to develop social and economic support to improve diet quality in individuals residing in highly deprived areas.

## Figures and Tables

**Figure 1 nutrients-11-02194-f001:**
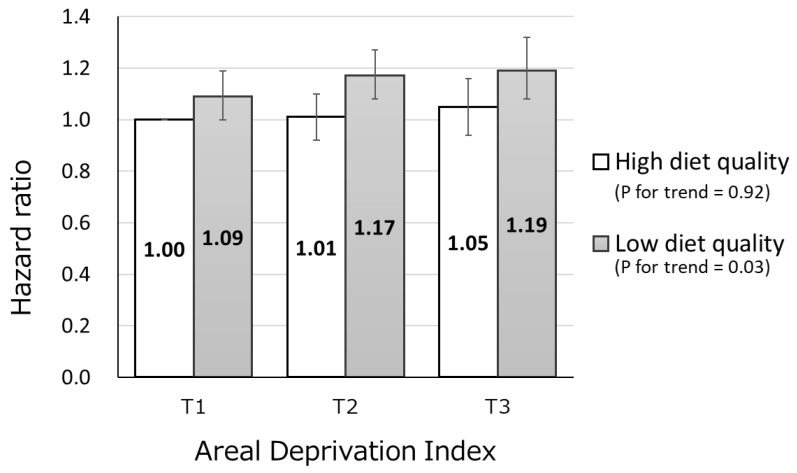
Multivariate-adjusted hazard ratios (95% confidence interval) of all causes of mortality, according to combinations of area deprivation index and diet quality.

**Table 1 nutrients-11-02194-t001:** Characteristics of subjects according to categories of deprivation index at the five-year follow-up survey.

	Area Deprivation Index		
	Tertile 1 (Low)	Tertile 2	Tertile 3 (High)
*N*	20,522	20,314	20,431
Area Deprivation index *	165.8–514.3	514.3–589.0	589.1–983.3
Population density	1640 (12.5)	1067 (12.5)	1173 (12.5)
Age (years) ^†^	51.1 (7.74)	50.8 (7.33)	52.2 (7.92)
Men (%)	46.8	45.5	44.8
Body mass index (kg/m^2^) ^‡, ¶^	23.3 (0.02)	23.5 (0.02)	24.0 (0.02)
Total physical activity (metabolic equivalents hours/week) ^‡, ¶^	32.9 (0.05)	33.4 (0.05)	33.8 (0.05)
Current smoker (%) ^¶, §^	16.8	15.8	12.2
Alcohol consumption ≥1 d/wk (%) ^¶, §^	35.1	32.0	23.1
History of hypertension (%) ^§^	17.8	16.6	17.3
History of dyslipidemia (%) ^§^	5.5	4.8	3.2
History of diabetes (%) ^§^	5.7	5.8	5.7
Occupation (agriculture, forestry and fishery, %) ^§^	13.8	20.4	26.6
Living alone (%) ^§^	2.8	3.4	4.4

* Range. ^†^ Mean (standard deviation). ^‡^ Age- and sex-adjusted means (standard error). ^¶^ Subjects with missing information were excluded (body mass index: *n* = 974; total physical activity: *n* = 9184; smoking status: *n* = 2089; alcohol consumption: *n* = 524). ^§^ Age- and sex-adjusted proportions.

**Table 2 nutrients-11-02194-t002:** Age- and sex-adjusted mean intakes of energy, nutrients, and food groups and the Japanese Food Guide Spinning Top score according to categories of deprivation index.

	Age- and Sex-Adjusted Mean (Standard Error)			*p* Trend *
	Area Deprivation Index			
	Tertile 1 (Low)	Tertile 2	Tertile 3 (High)	
Energy (kcal)	2034 (4.1)	2070 (4.1)	1951 (4.1)	<0.0001
Nutrient intake (/1000kcal)				
Total protein (g)	36.9 (0.04)	36.7 (0.04)	35.1 (0.04)	<0.0001
Total fat (g)	27.6 (0.05)	27.7 (0.05)	29.8 (0.05)	<0.0001
Saturated fat (g)	8.3 (0.02)	8.3 (0.02)	9.2 (0.02)	<0.0001
Monounsaturated fat (g)	9.5 (0.02)	9.5 (0.02)	10.4 (0.02)	<0.0001
Polyunsaturated fat (g)	6.4 (0.01)	6.4 (0.01)	6.7 (0.01)	<0.0001
Cholesterol (mg)	146 (0.5)	149 (0.5)	143 (0.5)	<0.0001
Total carbohydrate (g)	137 (0.2)	138 (0.2)	134 (0.2)	<0.0001
Dietary fiber (g)	6.9 (0.01)	6.7 (0.02)	6.3 (0.01)	<0.0001
Calcium (mg)	272 (0.7)	269 (0.7)	252 (0.7)	<0.0001
Sodium (mg)	2502 (13.1)	2452 (13.2)	2420 (13.2)	<0.0001
Food intake (g)				
Grains	550.1 (1.17)	571.9 (1.17)	527.4 (1.17)	<0.0001
Potatoes	30.1 (0.18)	29.0 (0.18)	24.5 (0.18)	<0.0001
Vegetables	226.0 (0.99)	227.7 (1.00)	222.0 (0.99)	0.005
Green and yellow vegetables	102.6 (0.54)	103.9 (0.54)	98.4 (0.54)	<0.0001
Other vegetables	123.4 (0.58)	123.8 (0.58)	123.6 (0.58)	0.78
Pickled vegetables	44.2 (0.29)	41.4 (0.29)	23.4 (0.29)	<0.0001
Fruit	243.3 (1.20)	238.2 (1.21)	185.9 (1.20)	<0.0001
Mushroom	11.5 (0.08)	11.6 (0.08)	8.1 (0.08)	<0.0001
Pulses	89.4 (0.60)	91.4 (0.60)	94.0 (0.60)	<0.0001
Fish and shellfish	100.8 (0.45)	103.7 (0.45)	81.5 (0.45)	<0.0001
Meat	60.7 (0.38)	62.4 (0.38)	71.6 (0.38)	<0.0001
Eggs	29.7 (0.22)	31.8 (0.22)	30.0 (0.22)	0.24
Dairy products	191.1 (1.34)	192.0 (1.35)	167.5 (1.35)	<0.0001
Coffee (≥1 cup/day, %)	28.5	32.3	36.3	<0.0001
Green tea (≥1 cup/day, %)	71.4	62.9	45.3	<0.0001
Japanese Food Guide Spinning Top score	48.3 (0.06)	48.5 (0.06)	46.6 (0.06)	<0.0001

* Adjusted for age and sex.

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
