# Peer review of "Diet Quality Affects the Association between Census-Based Neighborhood Deprivation and All-Cause Mortality in Japanese Men and Women: The Japan Public Health Center-Based Prospective Study"

_nutrients, 2019, doi:10.3390/nu11092194_

Round 1

Reviewer 1 Report

The authors present a population-based prospective cohort study examining the influence of diet quality on the relationship between neighborhood deprivation and mortality risk. The study features an impressive 16.5 year mean follow-up period and a large sample size. The data are very interesting, and I appreciate the opportunity to review it. However, there are a few limitations that should be addressed prior to publication. Most importantly, the language that the authors use to describe the research question is unclear. Are the authors testing whether or not diet quality is a moderator between the independent and dependent variable? If so, they should use this term so that the reader can easily understand the purpose of the study. Other minor suggestions are detailed below by section.

Introduction:

Lines 42 and 45: The specific mentions of the authors' previous work seem unnecessary. The reader can view the citations to understand the work is connected to the authors. Line 50: Suggest replacing might with may. Also, you mention diet quality first in this sentence, but then define it in the next paragraph. I suggest moving the definition to where you first mention the term. Lines 51-61: These paragraph seemed disjointed to me. In particular, lines 56-57 seemed out of place. It is repetitive of the first sentence of the paragraph. Yet, it is in the middle of the paragraph. Lines 64-65: Here is where you could introduce the concept of moderation.

Methods:

Lines 110 and 111: I suggest that you indicate that the FFQ has been validated in your specific population. This is a strength that warrants mentioning. It was confusing to me to go straight from data collection to the diet quality analyses (Spining Top Score derivatiton) in line 111. Perhaps it would be more clear if you began a new paragraph for this concept. Also, there is no subheading in this section for statistical or data analyses. Perhaps the study subjects information should be provided earlier so that the analyses and metrics can be discussed more seamlessly. From my American perspective, the occupation categories in lines 179-181) seem quite broad. Perhaps I am not fully appreciating the uniqueness of the categories. But, since the occupations are used as a proxy for SES, they are even more important. Do you have any idea of the years of education associated with each category? Given the relationship between education and life expectancy, it would be ideal if the occupation categories were organized in a way that captured differences in education. In particular, it seems like the housework/unemployed category could encompass a variety of education levels. As I understand it, there is a big daycare shortage in Japan that requires women in particular to leave their careers. In what occupation category would educated women who do not work outside the home fall?

Discussion:

1. There seems to be some repetition between the first and second paragraphs. I suggest writing these paragraphs more concisely.

Author Response

Manuscript ID: nutrients-572292

Title: Diet quality affects the association between census-based neighborhood deprivation and all-cause mortality in Japanese men and women: the Japan Public Health Center-based Prospective Study

Reviewer 1

The authors present a population-based prospective cohort study examining the influence of diet quality on the relationship between neighborhood deprivation and mortality risk. The study features an impressive 16.5 year mean follow-up period and a large sample size. The data are very interesting, and I appreciate the opportunity to review it. However, there are a few limitations that should be addressed prior to publication. Most importantly, the language that the authors use to describe the research question is unclear. Are the authors testing whether or not diet quality is a moderator between the independent and dependent variable? If so, they should use this term so that the reader can easily understand the purpose of the study. Other minor suggestions are detailed below by section.

[Response]

We thank the reviewers for their insightful comments on our paper. We have revised the manuscript by addressing each comment point-by-point as shown below, with all amendments in the text highlighted using a red font. We hope these changes will resolve any confusion and remedy the shortcomings of the paper.

In this study, we tested whether diet quality is a moderator between census-based neighborhood deprivation and all-cause mortality. We used this term in the revised manuscript (line 58).

Introduction:

Lines 42 and 45: The specific mentions of the authors' previous work seem unnecessary. The reader can view the citations to understand the work is connected to the authors.

[Response]

Thank you for your suggestion. We deleted “including our previous study” in line 44 and “our study in” in line 46.

Line 50: Suggest replacing might with may. Also, you mention diet quality first in this sentence, but then define it in the next paragraph. I suggest moving the definition to where you first mention the term.

[Response]

Thank you for your helpful suggestion. We replaced “might” with “may” in line 51. We also described the definition of lower diet quality in line 51-52 instead of the next paragraph.

Line 50-52 “Thus, people residing in neighborhoods within more deprived areas may have a lower diet quality, which is a lower adherence to the dietary guidelines, and be at higher risk of mortality.”

Lines 51-61: These paragraph seemed disjointed to me. In particular, lines 56-57 seemed out of place. It is repetitive of the first sentence of the paragraph. Yet, it is in the middle of the paragraph.

[Response]

Thank you for your helpful suggestion. We deleted the description of our previous study (in line 56-57 in the original manuscript), and moved the explanation of the Japanese Food Guide Spinning Top (in line 52-55 in the original manuscript) to the end of the introduction section in line 64-66 in the revised manuscript. We revised the second paragraph in the Introduction section as follows.

Line 53-57 “Several studies showed that individuals with a lower diet quality had a higher risk of mortality [10-13]. According to the report from the Global Burden of Diseases, unhealthy diet was an important ‘modifiable’ behavioral risk factor for many health conditions [14]. Improvement of diet quality might play a role in reduction of health disparities resulting from neighborhoods deprivation level.”

We revised the last sentence in the Introduction section as follows.

Line 62-66 “Here, we prospectively examined the association between deprivation in an area and all-cause mortality risk according to diet quality with reference to the Japanese Food Guide Spinning Top, which was jointly developed by the Ministry of Health, Labour, and Welfare and the Ministry of Agriculture, Forestry, and Fisheries [15].”

Lines 64-65: Here is where you could introduce the concept of moderation.

[Response]

According to your suggestion, we changed the sentence as follows.

Line 58-60 “A diet quality of individuals differs even if those who reside in the same deprived area. We hypothesized that diet quality as a moderator of the association between area deprivation and mortality.”

Methods:

Lines 110 and 111: I suggest that you indicate that the FFQ has been validated in your specific population. This is a strength that warrants mentioning. It was confusing to me to go straight from data collection to the diet quality analyses (Spining Top Score derivatiton) in line 111. Perhaps it would be more clear if you began a new paragraph for this concept.

[Response]

Thank you for your thoughtful suggestion. In this revision, we began a new paragraph for the diet quality analyses after the explanation of the FFQ.

Line 108-109 “…The food frequency questionnaire has reasonable validity and reproducibility [22-24].

We referred to the Japanese Food Guide Spinning Top, a chart designed for the general public indicating recommended daily servings …”

Also, there is no subheading in this section for statistical or data analyses.

[Response]

We already showed the subheading of “Statistical analysis” line 160.

Perhaps the study subjects information should be provided earlier so that the analyses and metrics can be discussed more seamlessly.

[Response]

We understood your suggestion. However, we excluded subjects according to their intakes of each dish category, so the definition of the dish categories should be shown before the description of the study subjects. We could not move “Study subjects” section earlier.

From my American perspective, the occupation categories in lines 179-181) seem quite broad. Perhaps I am not fully appreciating the uniqueness of the categories. But, since the occupations are used as a proxy for SES, they are even more important. Do you have any idea of the years of education associated with each category? Given the relationship between education and life expectancy, it would be ideal if the occupation categories were organized in a way that captured differences in education. In particular, it seems like the housework/unemployed category could encompass a variety of education levels. As I understand it, there is a big daycare shortage in Japan that requires women in particular to leave their careers. In what occupation category would educated women who do not work outside the home fall?

[Response]

Thank you for your thoughtful comment. We followed the definition of the occupation categories of our previous study (Kurotani et al. BMJ 2016; 352), which referred to the Standard Occupational Classification for Japan and the number of subjects. As for the education level, unfortunately, there are data on education for limited subjects only. Then, we could not consider the education in this study. This issue was described in the limitation section (line 307-311).

Line 307-311 “Finally, the effects of confounding by residual and unmeasured variables cannot be completely ruled out. Unfortunately, we did not have sufficient information on individual-level socioeconomic conditions. Specifically, we adjusted for each participant’s occupation in a broadly-defined category, which cannot precisely capture individual socioeconomic status.”

Regarding your question, we introduce our previous study using a part of this study subjects showing that high academic qualifications without an appropriate job could be a risk factor for stroke among Japanese women (Honjo et al. Stroke. 2014;45:2592-2598). The status inconsistency between occupation and education status might be related to the risk of health outcomes.

Discussion:

There seems to be some repetition between the first and second paragraphs. I suggest writing these paragraphs more concisely.

[Response]

Thank you for your suggestion. We deleted some repetition between the first and second paragraphs. We revised the first sentence of the second paragraph as follows (line 254-256).

Line 254-256 “We found that individuals residing in the more deprived area had not only low intakes of fruits and vegetables but also low intakes of fish and shellfish, and dairy products and high intake of meat.”

Reviewer 2 Report

Many thanks for the opportunity for revising this interesting paper. This study examines the associations between neighborhood deprivation and all-cause mortality risk according to diet quality. Authors found that a balanced diet might prevent the effect of living in a deprived neighbourhood on early death. Moreover, in individuals with high diet quality, no significant associations were found between area deprivation and mortality.

The manuscript is well written, the topic is relevant and scientifically sound, presents interesting findings and its contribution is valuable. Therefore, I found the manuscript suitable for publication. I will just make some minor comments for the authors to consider, I hope they found them useful.

In the last paragraph of the introduction section (lines 62-70) authors stated that they hypothesized low quality diet being more strongly associate with health problems in more deprived areas (being the moderator neighborhood deprivation). However, the aim of the study was to explore the association between neighborhood deprivation and mortality according to diet (thus, being the moderator diet quality). I think that this paragraph can benefit from describing the hypothesis of this research and explaining better why was selected more clearly.

In methods, authors describe that participants were excluded in different steps from the initial sample. Why authors excluded 4668 participants who reported the upper/lower 1% of sex-specific intake of each category? There was conducted any analysis to test bias related to missing values?

In the discussion I will suggest to address more in depth the causes by which authors think that diet quality contributes to reduce the risk of mortality related to area deprivation. Living in a deprived area might have an impact on health through multiple mechanisms. In addition, results showed that individuals residing in the more deprived areas were more likely to have poor diet, however, they were also less likely to be a current smoker and have habitual alcohol consumption. I think that the discussion can be improved adding a suggestion for interpreting the main results.

Other minor details:

In lines 112-114, 184-186, 200-205, 225-233, 248-251 review letter’s size. Review some typos in lines 174 and 178. In line 213 “table 2” should be in the text format not written as a title. Review the use of capital letters in the journals names in the references.

Author Response

Manuscript ID: nutrients-572292

Title: Diet quality affects the association between census-based neighborhood deprivation and all-cause mortality in Japanese men and women: the Japan Public Health Center-based Prospective Study

Reviewer 2

Many thanks for the opportunity for revising this interesting paper. This study examines the associations between neighborhood deprivation and all-cause mortality risk according to diet quality. Authors found that a balanced diet might prevent the effect of living in a deprived neighbourhood on early death. Moreover, in individuals with high diet quality, no significant associations were found between area deprivation and mortality.

The manuscript is well written, the topic is relevant and scientifically sound, presents interesting findings and its contribution is valuable. Therefore, I found the manuscript suitable for publication. I will just make some minor comments for the authors to consider, I hope they found them useful.

[Response]

We thank the reviewers for their insightful comments on our paper. We have revised the manuscript by addressing each comment point-by-point as shown below, with all amendments in the text highlighted using a red font. We hope these changes will resolve any confusion and remedy the shortcomings of the paper.

In the last paragraph of the introduction section (lines 62-70) authors stated that they hypothesized low quality diet being more strongly associate with health problems in more deprived areas (being the moderator neighborhood deprivation). However, the aim of the study was to explore the association between neighborhood deprivation and mortality according to diet (thus, being the moderator diet quality). I think that this paragraph can benefit from describing the hypothesis of this research and explaining better why was selected more clearly.

[Response]

Thank you for your thoughtful suggestion. We revised the last paragraph of the Introduction section according to your comment as follows.

Line 58-66 “A diet quality of individuals differs even if those who reside in the same deprived area. We hypothesized that diet quality as a moderator of the association between area deprivation and mortality. To our knowledge, no study has examined the association between the level of deprivation in an area and mortality according to diet quality. Here, we prospectively examined the association between deprivation in an area and all-cause mortality risk according to diet quality with reference to the Japanese Food Guide Spinning Top, which was jointly developed by the Ministry of Health, Labour, and Welfare and the Ministry of Agriculture, Forestry, and Fisheries [15].”

In methods, authors describe that participants were excluded in different steps from the initial sample. Why authors excluded 4668 participants who reported the upper/lower 1% of sex-specific intake of each category?

[Response]

Thank you for your comment. We excluded those who reported the upper 1% of sex-specific intake of each category or either the upper or lower 1% of sex-specific energy intake because we thought that their answers were not reliable.

There was conducted any analysis to test bias related to missing values?

[Response]

Thank you for your comment. We conducted the multiple imputation analysis in this revision, and found the similar results to those using dummy variables (we showed in the original manuscript). We adopted the multiple imputation methods, and we revised the Methods, Result, and Discussion sections as follows.

Line 181-187 “For participants with missing data on BMI, smoking status, physical activity, coffee consumption, green tea consumption or occupation, we imputed the data using multivariate normal imputation (the SAS PROC MI procedure) with 5 rounds of multiple imputations by including all covariates, follow-up length and mortality status to account for missing data. Then, we combined the estimates from each imputed dataset using the Rubin's rules (the SAS PROC MIANALYZE procedure) [26].”

Ref 26: Rubin DB. Multiple imputation for nonresponse in surveys. New York: Wiley, 1987.

Line 233-241 “… the multivariate adjusted HRs (95% CI) were 1.09 (0.999-1.19), 1.17 (1.08-1.27), and 1.19 (1.08-1.32) for those residing in the lowest through the highest third of ADI, respectively. However, individuals with a high quality diet had no significant association between ADI and mortality (P =0.92 for trend, P = 0.26 for interaction with diet quality); the multivariate adjusted HRs (95% CI) were 1.00 (reference), 1.01 (0.93-1.10), and 1.05 (0.96-1.16) in those residing in the lowest through the highest third of ADI, respectively.”

Figure 1

Line 248 “19%”

In the discussion I will suggest to address more in depth the causes by which authors think that diet quality contributes to reduce the risk of mortality related to area deprivation. Living in a deprived area might have an impact on health through multiple mechanisms. In addition, results showed that individuals residing in the more deprived areas were more likely to have poor diet, however, they were also less likely to be a current smoker and have habitual alcohol consumption. I think that the discussion can be improved adding a suggestion for interpreting the main results.

[Response]

Thank you for your thoughtful comment. We added the discussion for interpreting the difference in the association between ADI and mortality by diet quality as follows.

Line 280-285 “Whereas the causes of the difference in the association between ADI and mortality by diet quality were unclear, the diet quality might reflect other risk factors. We found that individuals with lower quality diet tended to be current smokers and living alone. However, we adjusted for smoking and living status in main analysis, suggesting that differential association by diet quality may not be ascribed to these factors.”

Table Characteristics of subjects according to diet quality

(This is only for the Reviewers)

Diet quality

Low

High

n

28410

32857

Area Deprivation index (mean (SD))

570 (91)

558 (85)

Age (y, mean (SD))

50.9 (7.7)

51.9 (7.6)

Men (%)

46.2

45.3

Body mass index (kg/m2, mean (SD))

23.1 (3.1)

23.6 (3.0)

Total physical activity (metabolic equivalents-h/wk, mean (SD)),

33.2 (6.3)

33.4 (6.2)

Smoking status

 Non-smoker (%)

63.0

70.7

 Past smoker (%)

8.2

9.3

 Current smoker (%)

28.8

20.0

History of hypertension (%)

19.1

19.1

History of dyslipidemia (%)

4.5

5.8

History of diabetes (%)

6.0

6.6

Occupation (agriculture, forestry and fishery, %)

20.6

24.4

Living alone (%)

5.0

3.6

Other minor details:

In lines 112-114, 184-186, 200-205, 225-233, 248-251 review letter’s size.

[Response]

Thank you for your comment. We changed the font to “12pt” in the text. Incidentally, we used the Preprint service to create the Nutrients format automatically for the original manuscript.

Review some typos in lines 174 and 178.

[Response]

Thank you for your careful review. We deleted “with a consumption (<20 or ≥20 cigarettes/day)” in line 171 and “or” in 174.

In line 213 “table 2” should be in the text format not written as a title.

[Response]

According to your comment, we moved the “Table 2” to the end of the sentence.

Line 216-220 “Individuals residing in the most deprived area had the lowest scores on the adherence to the Japanese Food Guide Spinning Top; the age- and sex-adjusted mean (standard error) scores were 48.3 (0.06), 48.5 (0.06), and 46.6 (0.06), (P <0.0001 for trend) in individuals residing in the least deprived area through the most deprived area, respectively (Table 2).”

Review the use of capital letters in the journals names in the references.

[Response]

Thank you for your careful review. We corrected the journal names.

Additionally, we added 3 references (ref no. 25, 26, and 29), and replaced ref no.6 with another reference (Ito et al.).